# Suppressor mutations in Rpf2–Rrs1 or Rpl5 bypass the Cgr1 function for pre-ribosomal 5S RNP-rotation

Matthias Thoms [1], Valentin Mitterer[1], Lukas Kater[2], Laurent Falquet[3], Roland Beckmann [2], Dieter Kressler [3] & Ed Hurt [1]

During eukaryotic 60S biogenesis, the 5S RNP requires a large rotational movement to achieve its mature position. Cryo-EM of the Rix1-Rea1 pre-60S particle has revealed the post-rotation stage, in which a gently undulating α-helix corresponding to Cgr1 becomes wedged between Rsa4 and the relocated 5S RNP, but the purpose of this insertion was unknown. Here, we show that *cgr1* deletion in yeast causes a slow-growth phenotype and reversion of the pre-60S particle to the pre-rotation stage. However, spontaneous extragenic suppressors could be isolated, which restore growth and pre-60S biogenesis in the absence of Cgr1. Whole-genome sequencing reveals that the suppressor mutations map in the Rpf2–Rrs1 module and Rpl5, which together stabilize the unrotated stage of the 5S RNP. Thus, mutations in factors stabilizing the pre-rotation stage facilitate 5S RNP relocation upon deletion of Cgr1, but Cgr1 itself could stabilize the post-rotation stage.

[1] Biochemistry Centre, University of Heidelberg, Heidelberg 69120, Germany. [2] Gene Center, University of Munich, Munich 81377, Germany. [3] University of Fribourg and Swiss Institute of Bioinformatics, Fribourg 1700, Switzerland. These authors contributed equally: Matthias Thoms, Valentin Mitterer. Correspondence and requests for materials should be addressed to M.T. (email: matthias.thoms@bzh.uni-heidelberg.de) or to E.H. (email: ed.hurt@bzh.uni-heidelberg.de)

Eukaryotic ribosome synthesis is a complex and highly spatially and temporally coordinated process that requires the consecutive action of more than 200 *trans*-acting assembly factors to meet the enormous cellular demand for accurately assembled mature ribosomal subunits[1–5]. The biogenesis pathway starts in the nucleolus with RNA-polymerase-I-catalysed transcription of ribosomal DNA into a large 35S precursor rRNA, which, upon concomitant and hierarchical joining of ribosome assembly factors and ribosomal proteins, is embedded into the huge 90S particle[6–10]. Endonucleolytic cleavage of the 35S pre-RNA subsequently generates the pre-40S and pre-60S particles, which from that point on undergo individual maturation and quality-control steps to finally join again in the cytoplasm forming translation-competent ribosomes.

The large 60S ribosomal subunit is composed of three rRNA species (25S/28S, 5.8S and 5S rRNA) and 46 (in yeast) or 47 (in human) ribosomal proteins[11,12]. Once separated from the pre-40S particles, the first individual precursors of the 60S subunit are formed within the nucleolus. Upon binding of ribosomal proteins, the nucleolar pre-60S maturation pathway is initiated by the appearance of the 27SA$_2$ pre-rRNA that is further processed to the 27SB pre-rRNA. Concomitantly, the intertwined rRNA domains are shaped into the developing 60S core in a consecutive order in which first the solvent-exposed side, followed by the polypeptide exit tunnel (PET) and finally the inter-subunit side are formed[13–16]. At the stage of nucleolar maturation intermediates, the 5S ribonucleoprotein particle (5S RNP), consisting of the 5S rRNA and ribosomal proteins Rpl5 (also known as uL11) and Rpl11 (also known as uL18), is already recruited, and the characteristic pre-60S 'foot' structure surrounding the internal transcribed spacer 2 (ITS2) RNA fragment has already formed[14,17–20]. Crucial pre-60S remodelling events, such as the removal of the Erb1–Ytm1 complex by the AAA–ATPase Rea1, facilitate the transition of the particle to the nucleoplasm[14,21–23]. A hallmark structure on early nucleoplasmic maturation intermediates, isolated via Arx1 or Nog2 (also known as Nug2), is the twisted 5S RNP, which adopts a conformation rotated ~180° compared to mature 60S subunits[18–20]. The recruitment of the Rix1 subcomplex, which allows stable docking of Rea1, and the removal of assembly factors Rpf2 and Rrs1 occur during the rotation of the 5S RNP into a near-mature conformation[20,24]. Subsequently, Rea1 performs its second restructuring role by triggering the release of Rsa4[25]. Prior to nuclear export, conformational proofreading of the particle takes places that links the removal of Rsa4 with activation and release of the GTPase Nog2, which in turn allows the recruitment of the export adaptor Nmd3[26]. After nuclear export, the AAA–ATPase Drg1 initiates the cytoplasmic maturation cascade by releasing the placeholder protein Rlp24, thus permitting the recruitment of Rpl24 (also known as eL24)[27,28]. Subsequent cytoplasmic pre-60S maturation steps include the Rei1–Jjj1–Ssa1-dependent dissociation of the export factor Arx1[29,30], assembly of the P-stalk and incorporation of Rpp0 (also known as uL10)[31,32], removal of Nmd3 by the GTPase Lsg1 coupled to the incorporation of Rpl10 (also known as uL16)[33–35], and release of the anti-association factor Tif6 promoted by Efl1 and Sdo1[30,36], which finally activates the 60S subunit to enter the pool of functionally translating ribosomes.

Whereas the 35S pre-rRNA is the common precursor of three of the four rRNA species (18S, 5.8S, 25S/28S), the 5S rRNA precursor is transcribed separately by RNA polymerase III. The 5S rRNA subsequently associates with the ribosomal proteins Rpl5 and Rpl11 to form the 5S RNP that is incorporated as a prefabricated complex adopting an immature conformation on the pre-60S particle[17,19]. Nuclear import of Rpl5 and Rpl11 is coordinated by the adaptor protein Syo1, which, in a second function, chaperones the 5S RNP until its pre-ribosomal assembly

by shielding exposed RNA-binding sites on Rpl11[37,38]. In addition, the heterodimer Rpf2–Rrs1 is thought to guide 5S RNP incorporation by providing a docking platform that anchors the 5S RNP in a network of interactions around the central protuberance (CP) involving the 25S rRNA and assembly factor Rsa4[39–41]. Therefore, the Rpf2–Rrs1 complex has to dissociate from the pre-60S particle, a reaction that appears to be necessary for 5S RNP relocation. However, to date, the mechanistic details of the events that trigger 5S RNP rotation have remained unexplored.

Here, we show that the small and conserved protein Cgr1, which was implicated in 60S biogenesis[20,42,43], plays a role in the relocation of the 5S RNP during 60S biogenesis. We found that yeast cells with a chromosomal *cgr1* deletion (*cgr1Δ*)—resulting in a slow-growth phenotype—exhibit a 5S RNP maturation defect on pre-60S particles. However, specific suppressor mutations could be isolated that map in genes encoding Rpf2, its binding partner Rrs1, and the ribosomal protein Rpl5. Owing to the nature of these suppressor mutations, which bypass Cgr1's function in this process, we were able to gain insight into the mechanism of 5S RNP rotation, revealing how untying of the twisted 5S RNP from its surrounding assembly factor network can drive 5S RNP rotation.

## Results

**Cgr1 marks pre-60S particles during 5S RNP rotation.** Cryo-EM analysis of the Rix1–Rea1 pre-60S particle showed that the 5S RNP had already rotated ~180° to its near-mature position[24], whereas in the 'upstream' pre-60S particles, such as the early Arx1 particle or Nog2 particle, the 5S RNP was still in the unrotated topology[19,20]. Among the many other structural peculiarities, the Rix1–Rea1 particle exhibited a 114 Å long, slightly undulating, α-helix inserted between the β-propeller domain of Rsa4 and the rotated 5S RNP, thereby clamping H38 of the 25S rRNA (A-site finger) at a new position (Fig. 1a)[24]. We suspected that this α-helix corresponds to the small, 120-amino-acid-long protein Cgr1 (Fig. 1b), which has been suggested to perform a role in pre-60S biogenesis[42–44]. Consistent with this interpretation, Gao and colleagues identified this long α-helix as Cgr1 in the early (unrotated 5S RNP) and late (rotated 5S RNP) states of their Nog2 pre-60S particles that resemble the early Arx1 and Rix1-Rea1 particles, respectively[20].

To find out with which pre-60S particles Cgr1 interacts, we first affinity purified both N- and C-terminally tagged Cgr1 from whole yeast cell lysates via TAP–Flag or Flag–TEV–ProtA (FTpA), respectively. Consistent with a predominantly nucleolar/nuclear localization of GFP–Cgr1 (Fig. 1c), the two different Cgr1 purifications were co-enriched for ribosome assembly factors that are typically present on intermediate pre-60S particles (i.e. Nog2, Rix1 and Arx1), and, accordingly, Cgr1 was not found on early nuclear (Ssf1 and Nsa1) or later cytoplasmic (Lsg1) particles (Fig. 1d, e, Supplementary Fig. 1a, b).

**Cgr1 depletion stalls the pre-60S prior to 5S RNP rotation.** To study the in vivo role of *CGR1* during 60S maturation, a *cgr1Δ* null strain was generated. Earlier data indicated that *CGR1* is either essential or non-essential for cell growth, depending on the strain background[42,43]. In our laboratory yeast strain, W303[45], *CGR1* is a non-essential gene, but displays an extreme slow-growth phenotype at all tested temperatures (23, 30 and 37 °C) (Fig. 2a). To analyse such a near-essential phenotype in a controlled way, we generated an auxin-inducible degron (AID)[46] allele of *CGR1*, which efficiently targeted Cgr1 for proteasomal degradation within 30–45 min of auxin addition (Supplementary Fig. 2a). This *CGR1*–HA–AID strain did not display an obvious

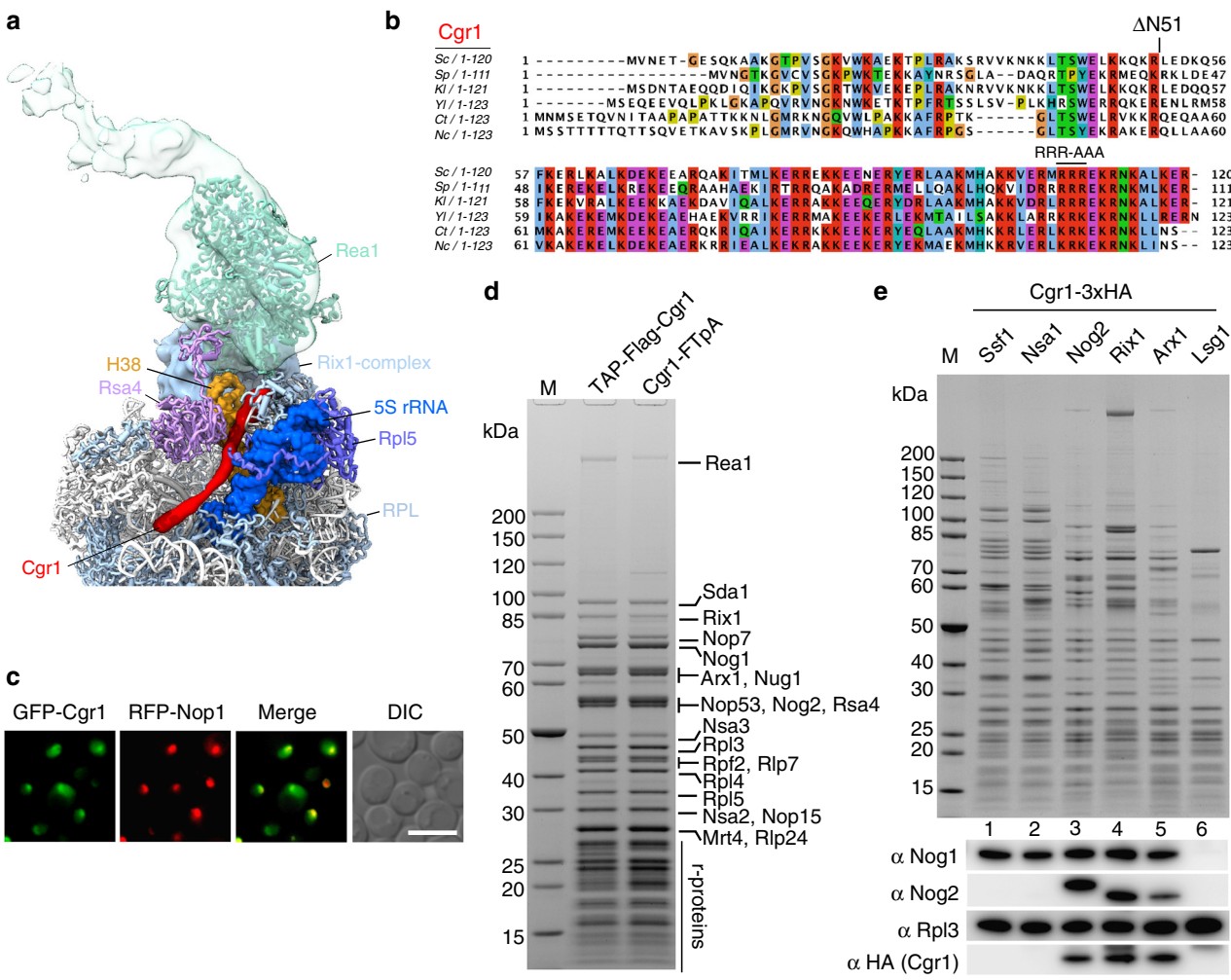

**Fig. 1** The short α-helical protein Cgr1 is wedged on nucleoplasmic pre-60S particles close to the rotated 5S RNP. **a** Cryo-EM position of Cgr1 wedged between the β-propeller domain of Rsa4 and the rotated 5S RNP on the Rix1-Rea1 pre-ribosomal particle (PDB: 5jcs,[24]). The 5S rRNA (blue) and H38 of the 25S rRNA (orange) are shown as a surface models, Cgr1 (red), Rsa4 (purple), Rea1 (cyan), Rix1-complex (light blue), Rpl5 (dark blue), and other ribosomal proteins (RPL, light blue) are depicted. **b** Multiple sequence alignment of Cgr1 orthologues from different fungal species: *Saccharomyces cerevisiae* (*Sc*), *Schizosaccharomyces pombe* (*Sp*), *Kluyveromyces lactis* (*Kl*), *Yarrowia lipolytica* (*Yl*), *Chaetomium thermophilum* (*Ct*) and *Neurospora crassa* (*Nc*); for the sequence alignment with higher eukaryotic orthologues including human Cgr1, see Supplementary Fig. 9. Two mutant constructs, Cgr1ΔN51 and Cgr1RRR > AAA, used for genetic interaction studies, are indicated above the alignment. **c** Subcellular distribution in yeast cells of GFP-tagged Cgr1 and RFP-Nop1 was monitored by fluorescence microscopy. The localization of GFP–Cgr1 is distributed over the nucleus, with the tendency to show a slightly stronger signal in the nucleolus. Scale bar is 5 μm. **d**, **e** Cgr1 is co-enriched on intermediate pre-60S particles typically found in the nucleus. **d** Cgr1 tagged either N-terminally (TAP-Flag) or C-terminally (FTpA) were isolated from yeast lysates in two affinity-purification steps. The final Flag eluates were analysed by SDS-PAGE followed by Coomassie staining. The bands identified by mass spectrometry are indicated. **e** Different pre-60S particles affinity purified via bait proteins Ssf1-FTpA (early nucleolar), Nsa1-FTpA (early nucleolar), Nog2–FTpA (intermediate nucleoplasmic), Rix1–FTpA (intermediate nucleoplasmic), Arx1-FTpA (intermediate nucleoplasmic to late cytoplasmic) and Lsg1-FTpA (late cytoplasmic) were affinity purified from yeast strains, which expressed Cgr1 carrying 3xHA (Cgr1–3xHA). Final eluates were analysed by SDS-PAGE and Coomassie staining (upper panel) or western blotting, using the indicated antibodies detecting Nog1, Nog2, Rpl3 and Cgr1 (lower panels). M: molecular weight marker

growth defect when incubated in the absence of auxin (Supplementary Fig. 2b), but exhibited a very mild half-mer phenotype, which could be due to the HA–AID-tag at the C-terminus (Fig. 2b). However, the polysome profile of the cells after auxin-dependent Cgr1–HA–AID depletion showed a drastic increase of the half-mer phenotype, consistent with previous findings[42] and indicative of a severe 60S biogenesis defect (Fig. 2b). Moreover, robust nuclear accumulation of the 60S reporter Rpl25–GFP was observed upon Cgr1 depletion, suggesting that the 60S maturation defect occurs prior to nuclear export (Fig. 2c).

Next, we wished to find out where exactly Cgr1 participates in the nuclear pre-60S maturation pathway. Since Cgr1 is closely intertwined with the interaction network around the CP,

adopting considerably different conformations depending on the rotation state of the 5S RNP[20], we hypothesized that the protein could function at a maturation step during 5S RNP relocation. To assess whether 5S RNP maturation might be affected in absence of Cgr1, we compared the assembly factor profile of Arx1-derived pre-60S particles, isolated from non-depleted (−auxin) versus Cgr1-depleted (+auxin) cells (Fig. 2d). Since Arx1 is associated with a broad range of pre-60S intermediates, from nuclear to cytoplasmic particles[18,19,47], it can serve as a bait to define the stage of pre-60S arrest by biochemical means. To allow monitoring of the 5S RNP maturation stage of the isolated particles, we used a strain expressing a chromosomal Rpf2–3xHA fusion, which is

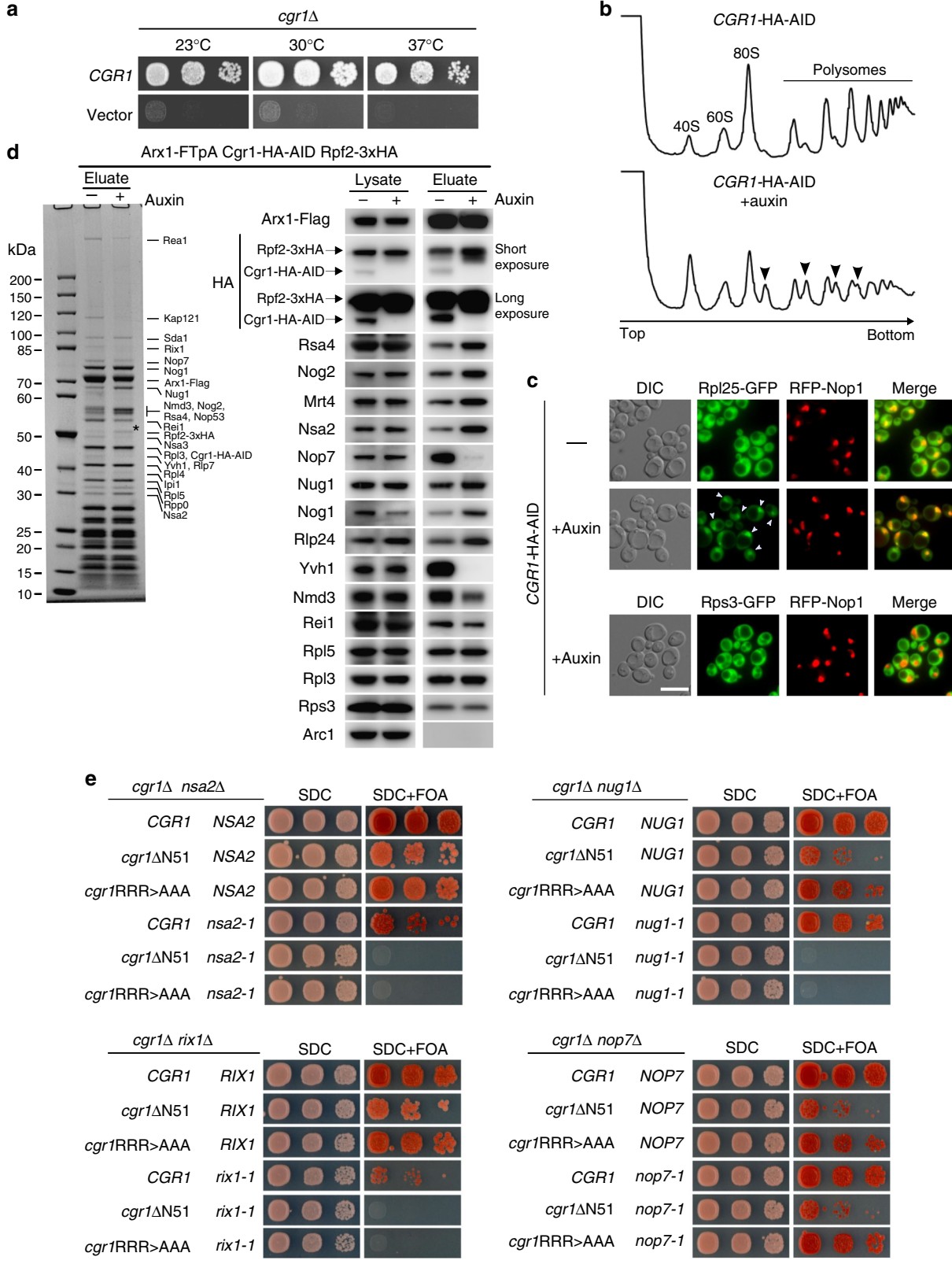

functional based on growth (Supplementary Fig. 1c), as it was suggested that the presence of the assembly factor Rpf2 in complex with its binding partner Rrs1 hinders 5S RNP rotation[17,20,39–41]. Indeed, western blot analyses revealed that Rpf2–3xHA became significantly enriched on Arx1 particles isolated from Cgr1-depleted cells in comparison to non-depleted

cells (Fig. 2d), indicating Cgr1 might facilitate 5S RNP relocation. Consistent with this finding, Cgr1 depletion caused a significant reduction of late-acting pre-60S factors (e.g. Yvh1, Rei1, Nmd3), whereas earlier assembly factors (e.g. Rsa4, Nog2, Nsa2, Mrt4, Rlp24) became more enriched (Fig. 2d). In contrast, the foot factors Nop7 and Nsa3 (also known as Cic1) were reduced on

**Fig. 2** Cgr1 plays a crucial role in ribosome biogenesis of pre-60S particles. **a** Chromosomal *CGR1* deletion in wild-type yeast strain W303 yields viable cells with an extreme slow-growth phenotype. The *cgr1Δ* shuffle strains transformed with empty plasmid or plasmid carrying wild-type *CGR1* were shuffled on SDC + FOA plates, before representative colonies were spotted in 10-fold serial dilutions on YPD plates. They were grown at the indicated temperatures for 2 days. **b** Cgr1 depletion impairs 60 S subunit synthesis. Polysome-profiles of *CGR1*–HA-AID (i.e. Cgr1-depletion strain) were recorded for untreated or auxin-treated (for 120 min) cells. Arrows denote ribosomal half-mers, indicating a specific 60S biogenesis defect. **c** pre-60S export is inhibited in cells depleted of Cgr1. Subcellular localization of the 60S reporter Rpl25–GFP, the 40S reporter Rps3–GFP and the nucleolar marker RFP–Nop1 was analysed in untreated or auxin-treated (for 120 min) *CGR1*-HA-AID cells. Arrows indicate nuclear accumulation of Rpl25–GFP. Scale bar is 5 μm. **d** Depletion of Cgr1 shifts Arx1 pre-60S particles to the early pool typical for the unrotated 5S RNP. Arx1–FTpA particles were affinity purified from untreated or auxin-treated (for 120 min) *CGR1*–HA–AID cells expressing a chromosomally integrated *RPF2*–3xHA variant. Lysates serving as input for the purifications and final eluates were analysed by SDS-PAGE and Coomassie staining. Indicated bands were identified by mass spectrometry (left panel, asterisk indicates Rpf2-3xHA) and western blotting based on specific antibodies (right panel). Rpf2 carries a 3xHA tag, whereas only one HA epitope is fused to Cgr1, explaining the different signal intensities of the HA western blots. **e** Synthetic lethal relationship between *cgr1* mutant alleles and distinct pre-60S assembly factors. Double-shuffle strains of *cgr1Δ* in combination with *nsa2Δ*, *rix1Δ*, *nug1Δ* and *nop7Δ*, respectively, were co-transformed with indicated plasmid-based wild type and mutant constructs. Transformants were spotted in 10-fold serial dilutions and growth on SDC-Leu-Trp (SDC) and SDC + FOA plates at 30 °C was monitored after 2 and 6 days, respectively. The *cgr1*RRR > AAA and *cgr1*ΔN51 mutants are shown in Fig. 1b and Supplementary Fig. 3. Published mutant alleles *nsa2-1*, *rix1-1*, *nug1-1* and *nop7-1* are listed in Supplementary Table 3

Cgr1-depleted particles, indicating that ITS2 processing and removal of the foot structure could proceed uncoupled of 5S RNP rotation.

**Genetic interactions between *cgr1* and pre-60S factors**. Next, we performed genetic analyses to further elucidate the in vivo function of Cgr1. For this purpose, we generated 'milder' *cgr1* mutant alleles compared to the *cgr1* null by truncating either the N-terminus (*cgr1*ΔN51) or mutating a cluster of positively charged residues at the C-terminus (R108A, R109A, R110A, *cgr1*RRR > AAA). The latter motif contacts a part of the 5S rRNA in the pre-rotation state[20]. Both of these *cgr1* mutants grew well at 30 °C compared to the *cgr1*-null, but exhibited a temperature-sensitive phenotype at 37 °C (Supplementary Fig. 3a, b). Combining *cgr1*ΔN51 or *cgr1*RRR > AAA with mutant alleles of other pre-60S assembly factors revealed a synthetic lethal phenotype at 30 °C in the case of *rix1-1*, *nsa2-1* and *nug1-1*, but not with *nop7-1* (Fig. 2e). The Rix1 subcomplex is implicated in the initiation of 5S RNP rotation[24], and an α-helix in the Nug1 N-terminal domain is in direct contact with and the Nsa2 N-domain in close proximity to Cgr1[19,20], whereas Nop7 is located far away at the 'foot' of the pre-60S particle[20]. Thus, the observed genetic relationships correlate well with the biochemical and cryo-EM data, reinforcing Cgr1's role in 5S RNP relocation.

**Specific suppressor mutations bypass the function of Cgr1**. During the course of growing the *cgr1Δ* strain on plates, we consistently noticed a few fast-growing colonies in the high-cell-density streak-out, which among other possibilities could be spontaneous suppressors that bypass the requirement for *CGR1* (Fig. 3a). To further elaborate on this possibility, we performed clarifying genetic tests with these putative suppressors. First, we backcrossed a few of these suppressor strains to a haploid *cgr1Δ* strain of opposite mating type, which harboured wild-type *CGR1* on a *URA3*-containing plasmid. After sporulation and tetrad dissection, the four germinated *cgr1Δ* spores containing *URA3*-*CGR1* plasmid showed a $2^{+}$:$2^{-}$ segregation regarding slow versus fast growth on 5-fluoroorotic acid (5-FOA) plates (Fig. 3b). Apparently, the fast-growth-suppressor phenotype points to a single mutated gene locus responsible for the extragenic suppression.

This finding prompted us to perform whole-genome DNA sequencing of two selected suppressor strains that upon backcrossing showed a 2:2 segregation (see above). Strikingly, in both strains a single missense mutation (G227A and C84F) in the open reading frame of the *RPF2* gene was found. The G227A mutation

mapped to the conserved sigma-70-like motif found in all members of the Brix protein family[48], whereas the C84F mutation is found in a conserved region known to be involved in the interaction with Rrs1[40]. Thus, the identified mutations, together with the observed accumulation of Rpf2 on pre-ribosomes after Cgr1 depletion, establish a direct link between Cgr1 and 5 S RNP maturation.

To find out whether suppressor mutations in genes other than *RPF2* exist, we systematically analysed the remaining *cgr1Δ* suppressors in a different way. For this purpose, we expressed the wild-type allele of *RPF2* and other factors suspected to functionally interact with Cgr1 on the pre-60S particles (i.e. *RRS1*, *RPL5* and *RPL11*; all placed under *GAL1-10* control) in all 41 *cgr1Δ* suppressor strains (39 remained uncharacterized) and tested for reversion of the fast-growing phenotype. Strikingly, overexpression of *RPF2* changed 30 suppressors, *RRS1* six suppressors and *RPL5* five suppressors into a slow-growth phenotype, suggesting that all of our isolated suppressor strains were hit in only three genes (Fig. 3c). Cloning and DNA sequencing of these suppressor genes revealed single point mutations in *RPF2* (25 unique exchanges), *RRS1* (four unique exchanges) and *RPL5* (four unique exchanges) (Table 1).

To confirm that the cloned suppressor alleles behave like anticipated, double-shuffle strains were generated, in which *cgr1Δ* was finally combined with the given cloned suppressor allele. This genetic analysis revealed that all identified suppressor alleles complemented the severe growth defect of *cgr1Δ* mutant cells, although wild-type growth levels were not reached (Fig. 3d–f, Supplementary Fig. 4).

**cgr1Δ suppressor mutations within the pre-60S structure**. We sought to localize the suppressor mutations within the cryo-EM structure of pre-60S particles, where the 5S RNP is still unrotated and in direct contact with the Rpf2–Rrs1 heterodimer[19,20] (Fig. 4, Supplementary Figs. 5–7 and Table 1). For Rpf2, where a total of 25 different suppressor mutations were isolated, three mutations (A10E, R14I, K18T) map in an N-terminal α-helix interacting with H83 and H87 of 25S rRNA, whereas the remaining ones are distributed throughout the Brix-fold, which broadly participates in the interaction with both the 5S RNP and Rrs1 (Fig. 4, Supplementary Fig. 5). Several of these mutations showed substitutions of surface-exposed basic residues that change the electrostatic surface potential. Notably, surface-exposed basic amino acid clusters within Rpf2 were recently analysed in vitro, demonstrating that highly conserved R236, the R62–K63 cluster and the KKR loop (residues 94–96) are important for 5S rRNA binding[39]. Strikingly, the Rpf2 R62L/S, K63T and R236G/I

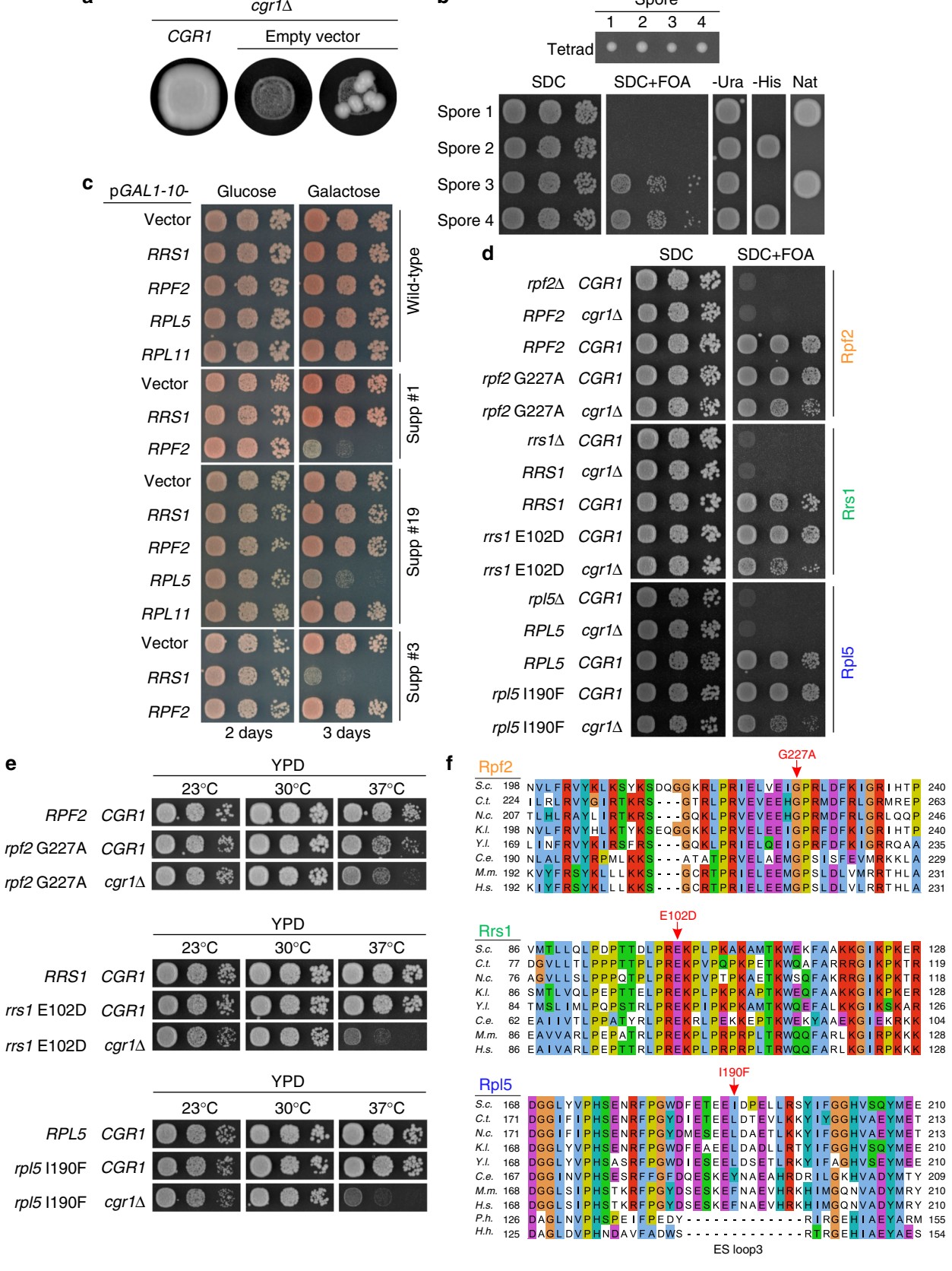

**Fig. 3** Suppressor mutations in *RPF2*, *RRS1* and *RPL5* bypass the requirement for *CGR1*. **a** Dot spot growth analyses of the *cgr1*Δ strain, harbouring plasmid-borne *CGR1* (left panel) or empty plasmid (middle and right panels), incubated on YPD plates at 30 °C for 3 days. The dot spot on the right, but not the middle, exhibits faster-growing colonies, which are suppressors of *cgr1*Δ. **b** *cgr1*Δ (*cgr1*::natNT2) suppressor strain was crossed with a *cgr1*Δ (*cgr1*:: *HIS3*MX6) strain containing *CGR1* on a *URA3* plasmid (pRS316-*CGR1*). After sporulation and tetrad dissection (upper panel shows a representative tetrad), the four haploid spores were tested for growth in the absence of pRS316-*CGR1* on SDC + FOA plates, for the presence of pRS316-*CGR1* on SDC-Ura, and for the presence of the *CGR1* gene disruption markers on SDC-His and YPD + clonNat. Cells were spotted in 10-fold serial dilutions and incubated at 30 °C for 2 days (lower panel). **c–f** Suppressor mutations are located in genes encoding *RPF2*, *RRS1* and *RPL5*. **c** Wild type and different *cgr1*Δ suppressor strains (suppressor #1, #3 and #19) were transformed with plasmids expressing *RPF2*, *RRS1*, *RPL5* or *RPL11* under the control of the galactose-inducible *GAL1-10* promoter. Representative transformants were spotted in 10-fold serial dilutions on SDC plates containing glucose (*GAL* repression) and galactose (*GAL* induction) and growth was assessed after incubation at 30 °C for 2 and 3 days, respectively. **d, e** Double-shuffle strains of *cgr1*Δ (+ p*URA3-CGR1*) combined with *rpf2*Δ (+ p*URA3-RPF2*), *rrs1*Δ (+ p*URA3-RRS1*) and *rpl5*Δ (+ p*URA3-RPL5*), respectively, were transformed with plasmids harbouring the suppressor allele or the respective wild-type gene combined with plasmids harbouring wild-type *CGR1* or empty plasmid. Transformants were spotted in 10-fold serial dilutions on SDC + FOA plates (**d**) and after plasmid shuffling on YPD plates (**e**). Growth was analysed after incubation for 2 days at the indicated temperatures. **f** Multiple sequence alignment of Rpf2, Rrs1 and Rpl5 orthologues from *Saccharomyces cerevisiae* (*S.c.*), *Chaetomium thermophilum* (*C.t.*), *Neurospora crassa* (*N.c.*), *Kluyveromyces lactis* (*K.l.*), *Yarrowia lipolytica* (*Y.l.*), *Caenorhabditis elegans* (*C.e.*), *Mus musculus* (*M.m.*), *Homo sapiens* (*H.s.*), *Pyrococcus horikoshii* (*P.h.*) and *Halobacterium hubeiense* (*H.h.*). The respective suppressor alleles analysed in **d** and **e** are indicated

## Table 1 Comparison of the isolated *cgr1*Δ suppressor mutations

| Protein | Point mutation | Interaction/role |
|---|---|---|
| Rpf2 (30 isolated suppressor strains) | A10E (2 strains) | 25S rRNA (CP H87) |
| | R14I (2 strains) | 25S rRNA (CP H87) |
| | K18T | 25S rRNA (CP H87) |
| | D48Y | Folding |
| | K53R | Folding |
| | K54E | 5S rRNA |
| | R62L | 5S rRNA/25S rRNA |
| | R62S | 5S rRNA/25S rRNA |
| | K63T | 5S rRNA/25S rRNA |
| | N64K | 5S rRNA/25S rRNA |
| | K81N | 25S rRNA |
| | K81T (2 strains) | 25S rRNA |
| | C84F | Folding |
| | C84W | Folding |
| | S93F | 5S rRNA (KKR loop) |
| | R104L | Rrs1, folding |
| | D112Y (3 strains) | Folding |
| | M117V | 5S rRNA (KKR loop), folding |
| | G177R | Rpl5/Rrs1, folding |
| | H180N | Folding |
| | V203F | Rrs1, folding |
| | G227A | Rrs1 (sigma70-like motif), folding |
| | G227V | Rrs1 (sigma70-like motif), folding |
| | R236G | 5S rRNA |
| | R236I | 5S rRNA |
| Rrs1 (6 isolated suppressor strains) | L92H | Rpf2 |
| | E102D | Rpf2 |
| | K103N (3 strains) | Rpf2 |
| | P106Q | Rpf2 |
| Rpl5 (5 isolated suppressor strains) | V73F (2 strains) | 5S rRNA/Rpf2 |
| | E126K | Rsa4/Rpf2 (ES loop 2) |
| | E128K | Rsa4/Rpf2 (ES loop 2) |
| | I190F | Rsa4/5S rRNA (ES loop 3) |

CP = central protuberance; ES = eukaryote-specific

mutations were all among our *cgr1*Δ suppressors. Although no mutations in the highly conserved KKR motif were found, the suppressor mutation Rpf2 S93F is within this KKR loop as well, which may be destabilized by the S93F change (Fig. 4, Supplementary Fig. 5), and hence could be the cause of a reduced interaction with the 5S rRNA. Consistent with this data, specific

point mutations in the 5S rRNA tip, mediating the interaction with the Rpf2 KKR loop, strongly impaired the interaction between Rpf2–Rrs1 and the 5S rRNA[41]. Other suppressor mutations in Rpf2, such as D48Y, D112Y or H180N, are found within the Brix-domain fold and eventually destabilize the Rpf2–Rrs1 interaction (Fig. 4, Supplementary Fig. 5).

In the case of Rrs1, all identified mutations are clustered in a highly conserved, proline-rich unstructured region (residues 92–108), which protrudes from the Rpf2 interaction-domain and continues into the carboxy-terminal sequence that contacts the 25S rRNA at multiple sites (Fig. 4, Supplementary Fig. 6), thereby also stabilizing the unrotated 5S RNP[20,40]. In vitro, both the proline-rich region and the C-terminal end of Rrs1 are not required for complex formation with Rpf2[39], but, in the cryo-EM structure, the proline-rich region is in contact with the Brix1-fold domain of Rpf2 (Fig. 4, Supplementary Fig. 6). Therefore, it is conceivable that our identified suppressor mutations in the proline-rich Rrs1 loop might change the position of the Rrs1 C-terminus, and thereby destabilize the unrotated 5S RNP.

Interestingly, three of our discovered suppressor mutations map to the ribosomal protein Rpl5 (E126K, E128K, I190F), specifically in two of the three eukaryote-specific loop regions required for 60S biogenesis, which if deleted cause trapping of Rpf2–Rrs1 on pre-60S particles[49]. In particular, the I190F mutation is located in eukaryotic-specific loop 3 (residues 185–198) that bridges Rpl5 with the Rsa4 β-propeller and the twisted 5S rRNA, whereas the mutations E126K and E128K map to the neighbouring eukaryote-specific loop 2 (residues 122–138), which is not resolved in the cryo-EM structure, but most likely connects the β-propeller of Rsa4 and Rpf2 (Fig. 4, Supplementary Fig. 7). In contrast, the fourth suppressor mutation within Rpl5 (V73F) maps to a conserved short loop motif sandwiched between the 5S rRNA and Rpf2 (Fig. 4, Supplementary Fig. 7).

Thus, considering all these different suppressor mutations in the structural context of the pre-60S particle, they likely destabilize the intricate interaction network between Rpf2–Rrs1, Rsa4 and the unrotated 5S RNP, which consequentially could allow driving the equilibrium towards the rotated state of the 5S RNP, thus compensating for the absence of Cgr1.

**Suppressor mutants promote 5S RNP rotation in *cgr1*Δ strains.** Based on the structural interpretation of the various *cgr1*Δ suppressors, we examined the impact of a few of these mutations on pre-60S maturation. First, we determined the localization of the 60S export reporter Rpl25–GFP in the mutants *rpf2*V203F, *rrs1*E102D and *rpl5*I190F, which all confer a strong suppression

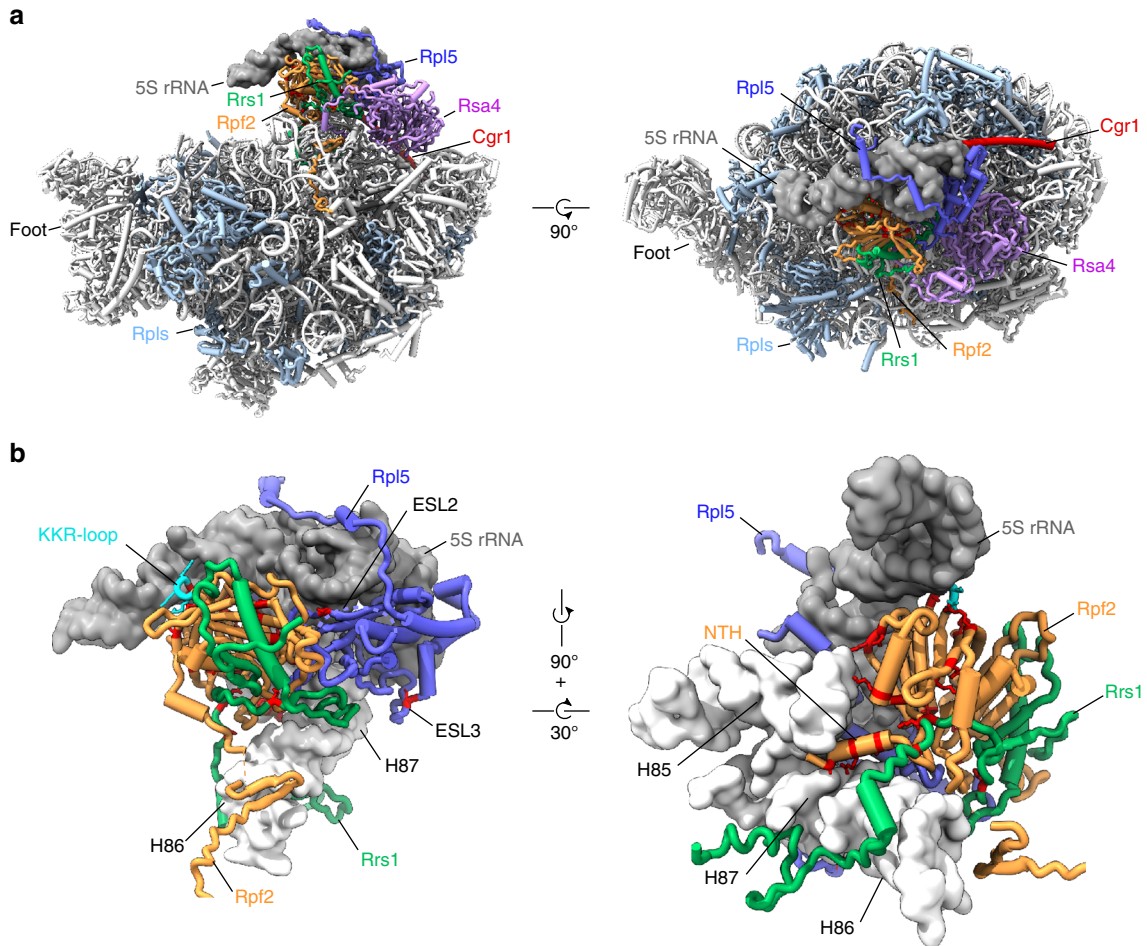

**Fig. 4** *cgr1Δ* suppressor mutations in *RPF2*, *RRS1* and *RPL5* destabilize the unrotated 5S RNP on the pre-60S particle. **a** Overview of biogenesis factors Rpf2 (orange), Rrs1 (green), Rsa4 (purple), Cgr1 (red) and the ribosomal protein Rpl5 (dark blue) on the Nog2 particle (PDB: 3jct,[20]) in the front and top view. Ribosomal proteins are shown in light blue and 5S rRNA as a dark grey surface model filtered at 6 Å resolution. **b** Positions of *cgr1Δ* null suppressor mutations (red) displayed with surface models of the 5S rRNA (dark grey) and helices H83 to H87 (nucleotides 2650–2754) of the 25S rRNA (light grey). The KKR-loop of Rpf2 is highlighted in cyan. ESL2 and ESL3 mark the eukaryotic-specific loops 2 and 3 of Rpl5, NTH marks the N-terminal helix of Rpf2

phenotype on *cgr1Δ*. In contrast to wild-type *RPF2*, *RRS1* and *RPL5* cells, which display nuclear accumulation of Rpl25–GFP after auxin-dependent Cgr1–HA–AID depletion, efficient nuclear export of Rpl25–GFP was re-established in the respective suppressor strains, which is clear-cut evidence for resuming pre-60S biogenesis (Fig. 5a). Moreover, we analysed the assembly factor composition of Arx1-affinity purified pre-60S particles derived from the *rpf2*V203F, *rrs1*E102D and *rpl5*I190F suppressor mutants, before and after Cgr1 depletion (Fig. 5b). In all cases, the pattern of factor enrichment on and removal from the Arx1 particles was consistent with our previous interpretation that nuclear export of pre-60S subunits was re-established in *cgr1Δ* cells by specific mutations in Rpf2, Rrs1 and Rpl5 (Fig. 5b). Notably, the assembly factor Rpf2, which became enriched on Arx1 pre-60S particles upon Cgr1 depletion (see also above), co-purified similar to the wild-type condition in the suppressor mutants (Fig. 5b). This finding further strengthens the hypothesis that the bypassing function of the suppressors could be specifically connected to a step in the course of 5S RNP relocation.

To directly assess whether 5S RNP rotation is inhibited in pre-60S particles when Cgr1 is depleted, but restored in the suppressor mutants, we performed cryo-EM analysis (Fig. 6, Supplementary Fig. 8 and Supplementary Table 1). This method showed that in the Cgr1 non-depleted strain (Arx1–FTpA Cgr1–

HA–AID, no auxin), which served as control, the 5S RNP was rotated in ~40% of the Arx1 particles, whereas ~60% of particles exhibited the non-rotated stage (Fig. 6a). This ratio is typical for the distribution of rotated (mature) versus non-rotated (immature) 5 S RNP in Arx1 or Nog2 pre-60S particles[19,20]. Strikingly, the 5S RNP remained to 100% non-rotated in the *cgr1*-depletion mutant (Fig. 6b; Arx1–FTpA Cgr1–HA–AID, + auxin). However, 5S RNP relocation was significantly restored in the *rrs1*E102D suppressor strain, showing 23% of the Arx1 pre-60S particles in the post-rotation stage (Fig. 6c; Arx1–FTpA Cgr1–HA–AID *rrs1*E102D, + auxin). Thus, structural analysis also supports the view that the suppressor mutations facilitate 5S RNP rotation in the absence of Cgr1, which explains well why the suppressor strains can re-export pre-60S particles and regain better cell growth. However, suppressor mutants did not reach optimal growth (see also Fig. 3e), which may correlate with the degree of the 5S RNP relocation. Notably, the cryo-EM analysis further revealed that the foot structure, carrying the ITS2 fragment of the 7S pre-rRNA and associated assembly factors, was absent from the Cgr1-depleted Arx1 particles (Fig. 6b). This finding is in line with the biochemical data demonstrating a strong decrease of foot factors Nop7 and Nsa3 on these particles (see Figs. 2d and 5b), which suggests that maturation of the foot can proceed independent of 5S RNP maturation.

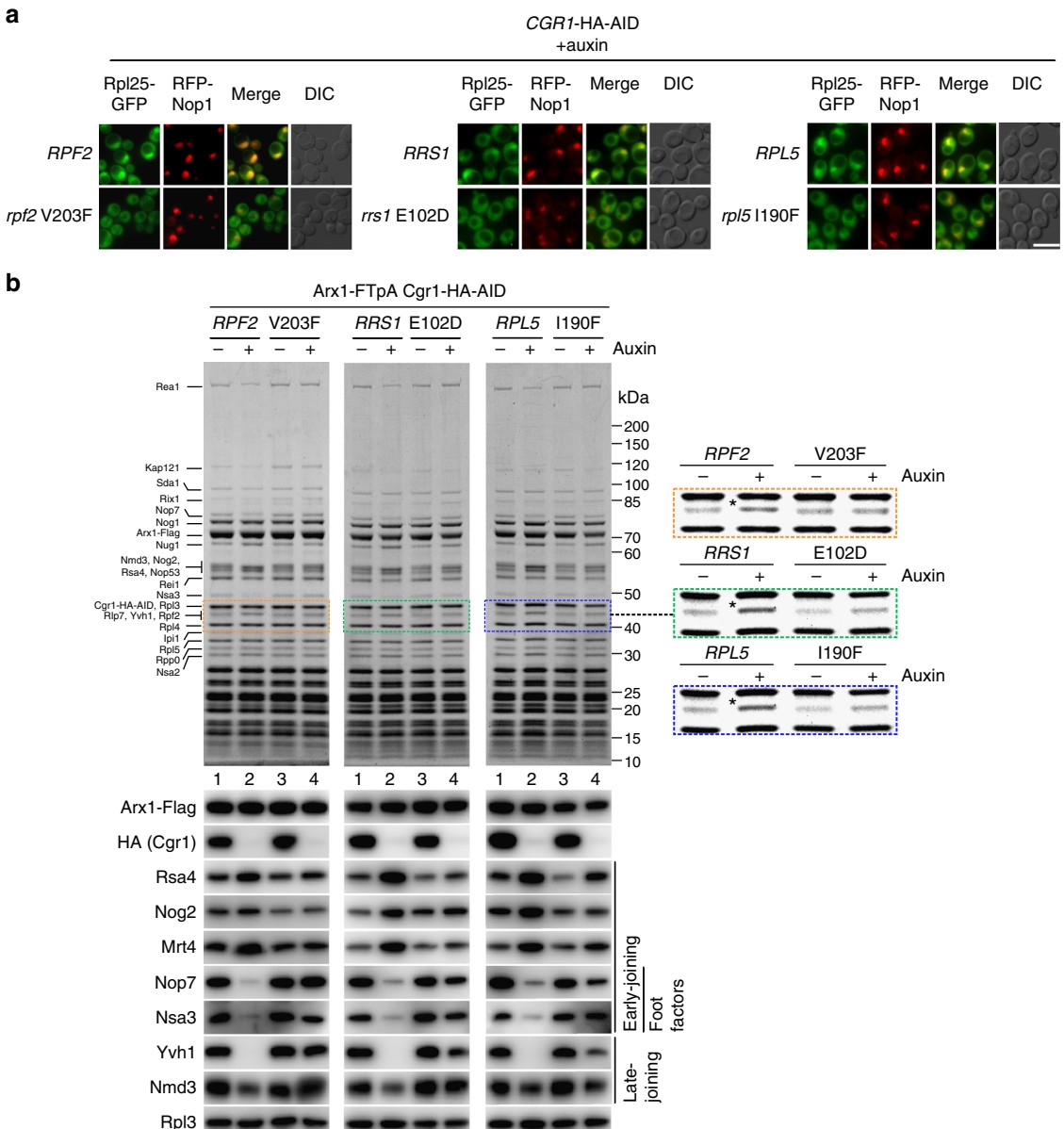

**Fig. 5** Suppressor mutations in Rpf2, Rrs1 and Rpl5 rescue the 60S biogenesis defect in Cgr1-depleted cells. **a** Nuclear pre-60S export is restored in suppressor mutants after Cgr1 depletion. Subcellular location of Rpl25–GFP and RFP–Nop1 (nucleolar marker) was examined in *CGR1*–HA–AID cells expressing either wild-type *RPF2*, *RRS1* and *RPL5* or the respective mutant alleles *rpf2*V203F, *rrs1*E102D and *rpl5*I190F, after incubation with auxin for 120 min. Scale bar is 5 μm. **b** Biochemical maturation of Arx1 pre-60S particles is restored in *cgr1Δ* suppressor mutants. Arx1–FTpA particles were affinity purified from *CGR1*–HA–AID cells expressing either wild-type *RPF2*, *RRS1* or *RPL5*, or the indicated suppressor mutants before and after treatment with auxin for 120 min. Final eluates were analysed by SDS-PAGE and Coomassie staining (indicated bands were identified by mass spectrometry; upper panels) or by western blotting using the antibodies shown on the left (lower panels). The area of the Coomassie-stained SDS-polyacrylamide gel to which Rpf2 migrates is enlarged on the right to better reveal how the intensity of co-enriched Rpf2 changes, depending on Cgr1 depletion in the various suppressor mutants

## Discussion

In this study, we unveiled a function of the small conserved α-helical protein Cgr1 (Supplementary Fig. 9) in 5S RNP rotation during 60S biogenesis, which occurs in the nucleus prior to nuclear export. Previous findings have shown that Cgr1 decorates nuclear pre-60S particles, which are in the process of 5S RNP rotation[20,24]. Due to its topological positioning, Cgr1 can ideally influence progression through this maturation step, by either affecting the transition stage to overcome the rotational block or by stabilizing the rotated stage. Consistent with this interpretation, pre-60S particles are shifted back to the pre-rotational stage

in *cgr1Δ* cells, thus identifying the arrest of 60S maturation as a possible cause of the severe slow-growth phenotype of the *cgr1*-null mutant. However, this defect can be overcome by second-site revertants (i.e. extragenic suppressor mutations), which allow resumption of cell growth. Strikingly, all the isolated suppressor mutations map in only three factors—Rpf2, Rrs1 and Rpl5—which normally under wild-type conditions keep the 5S RNP on pre-60S particles in the pre-rotation stage. Thus, the identified suppressor mutations hint to the mechanism by which cells can re-locate the 5S RNP during 60S biogenesis in the absence of Cgr1. Accordingly, depletion of Cgr1 results in inhibition of 5S

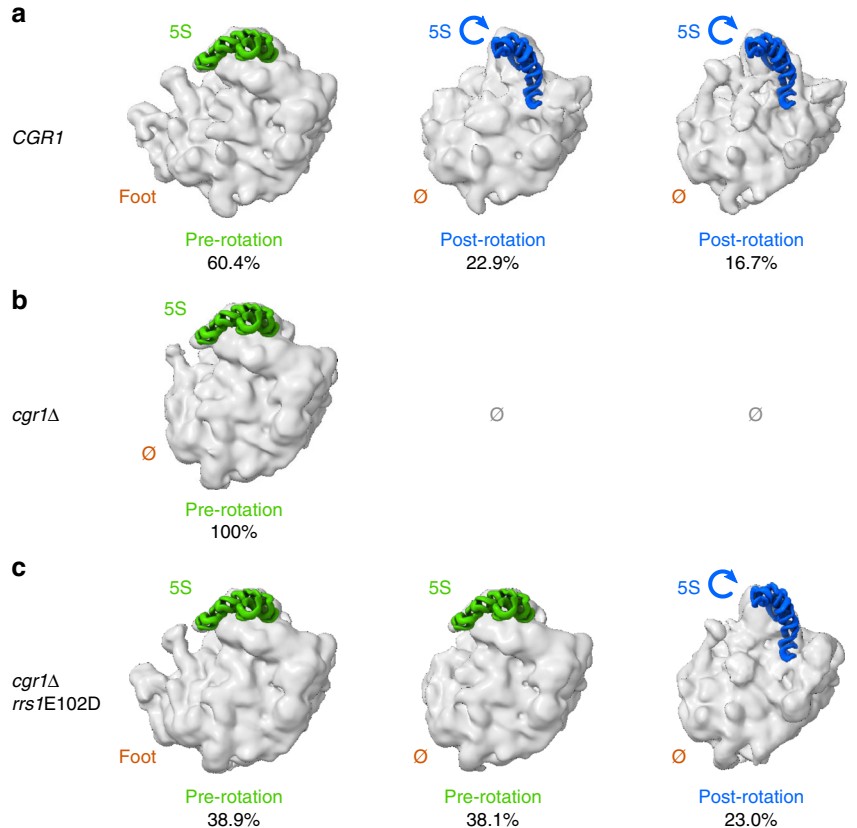

**Fig. 6** Cryo-EM reveals inhibition of 5S RNP rotation in *cgr1Δ* cells but restoration in *rrs1E102D* suppressor mutant. **a–c** 3D cryo-EM reconstructions of pre-60S particles affinity purified via the Arx1 bait protein from the indicated yeast strains. **a** *CGR1* control strain: *rrs1Δ* [YCplac111–*RRS1*] Arx1–FTpA Cgr1–HA–AID; -auxin. **b** *cgr1Δ* depleted: *rrs1Δ* [YCplac111–*RRS1*] Arx1–FTpA Cgr1–HA–AID; + auxin (2 h). **c** *cgr1Δ* depleted in the presence of the *rrs1E102D* suppressor: *rrs1Δ* [YCplac111–*rrs1E102D*] Arx1–FTpA Cgr1–HA–AID + auxin (2 h). For each obtained class of the respective data set, the rotation state of the 5S is indicated by a fit model of 5S rRNA taken from PDB: 3jct (green: pre-rotation,[20]) or PDB: 5jcs (blue: post-rotation,[24]). Also, the presence or absence of the ITS2-harbouring foot structure is indicated

RNP rotation, but suppressor mutations mapping in factors stabilizing the pre-rotational stage of the 5S RNP allow to partly overcome this defect.

As previously observed in the cryo-EM structure of the Rix1–Rea1 particle, the gently undulating C-terminal α-helix of Cgr1 is wedged between the rotated 5S RNP, the relocated A-site finger H38 and the β-propeller domain of Rsa4, thereby stabilizing the rotated 5S in a position that hinders back rotation[24]. In the 'early' state 1 of Nog2 pre-60S particles (resembling the 'early' pool of Arx1 pre-60S particles), in which the 5S RNP is non-rotated, the binding sites for Cgr1 are very different compared to those of the rotated stage[19,20]. Specifically, Cgr1 is located on the solvent side in the pre-rotation stage, contacting H38 as well as one tip of the unrotated 5S RNP, whereas after 5S RNP rotation, the Cgr1 α-helix adopts a more straightened conformation and is clamped between Rsa4 and the 5S RNP, thereby holding the relocated H38 in a bent position on the inter-subunit side. This rearranged topology suggests that Cgr1 accompanies or even facilitates H38 relocation from the solvent to the inter-subunit side. In addition, by snapping in after relocation, Cgr1 could stabilize the rotated 5S RNP position. It is tempting to speculate that upon initiation of the 5S RNP rotational movement, potentially induced by the recruitment of the Rix1 subcomplex[24], the contact between the 5S RNP and Cgr1's C-terminal α-helix gets temporarily disconnected, which could allow H38 to slide under the detached Cgr1 C-terminus. During the subsequent 5S RNP rotation, the straightened and co-rotating Cgr1 α-helix could continuously exert pressure on H38, which helps to bring it into the new

position at the inter-subunit interface. Interestingly, in bacteria, a key role for H38 in the maturation of the CP was postulated[50]. Depletion of the circularly permuted GTPase YlqF allowed the isolation of late ribosome assembly intermediates with an immature CP, which was highly disordered with no obvious structured intermediate, in contrast to the stable arrangement of the 5S RNP in the 'early' Arx1 particle. Nevertheless, it was shown that in these particles, H38 also adopts different orientations, and it was suggested that re-orientation of the A-site finger is a prerequisite for stable CP formation.

In summary, this study provided mechanistic insight into the 5S RNP rotation during large subunit biogenesis and its coupling to pre-60S nuclear export. This could be achieved by combining classical yeast genetic methods with modern whole-genome high-throughput sequencing, which appears to be an effective approach to further unravel the complicated process of eukaryotic ribosome assembly.

## Methods

**Yeast strains and plasmids.** The *Saccharomyces cerevisiae* strains used in this study were derived from W303[45] and are listed in Supplementary Table 2. Strains were constructed by using established gene disruption, genomic tagging[51,52], mating and tetrad dissection methods. Shuffle strains were constructed by knocking out an essential gene in a diploid yeast strain, transformation with a *URA3* plasmid containing the respective wild-type gene and sporulation to generate haploids harbouring the gene knockout and the complementing *URA3* plasmid. Subsequently, double-shuffle strains containing knockouts of *cgr1* and an essential gene (as indicated in the respective figures) complemented by two *URA3* plasmids harbouring the corresponding wild-type genes were generated by crossing

of two in the first step generated shuffle strains with opposing mating types and subsequent sporulation and identification of haploids containing both knockouts and both URA3 plasmids (i.e. spores containing both selection markers used for the two knockouts, fast-growing on plates lacking uracil, and non-viable on 5-FOA containing plates).

The plasmids used in this study are listed in Supplementary Table 3 and were constructed according to standard DNA cloning techniques and verified by sequencing.

**Identification of suppressors by high-throughput sequencing.** The two suppressor mutants and the CGR1 shuffle strain (parental control strain) were grown in YPD medium to an $OD_{600}$ value of around 1, and cells corresponding to 20 $OD_{600}$ units were harvested by centrifugation. Genomic DNA was extracted essentially as described in Current Protocols in Molecular Biology[53]. After washing in $dH_2O$, cells were transferred to a 2.2 ml safe-seal Eppendorf tube, centrifuged again and resuspended in 200 μl breaking buffer [100 mM NaCl, 10 mM Tris-HCl (pH 8), 1 mM EDTA (pH 8), 2% Triton X-100, 1% SDS]. After addition of 0.3 g glass beads and 200 μl phenol–chloroform–isoamyl alcohol (49.5:49.5:1; Sigma), cells were broken by vigorous vortexing for 3 min. Then, 200 μl of TE buffer [10 mM Tris-HCl (pH 7.5), 1 mM EDTA (pH 8)] was added and the tubes were briefly vortexed. Tubes were centrifuged for 5 min at 13,500 rpm in an Eppendorf centrifuge and the aqueous upper phase was transferred to a 1.5 ml Eppendorf tube. Then, 1 ml of absolute ethanol was added and the contents of the tubes were mixed by inversion. Following centrifugation for 3 min at 13,500 rpm, the supernatant was removed and the pellet was resuspended in 400 μl of TE buffer. To digest the RNA, 30 μl of a 1 mg/ml DNase-free RNase A solution (Sigma) was added and the tubes were incubated for 5 min at 37 °C. Genomic DNA was then precipitated upon addition of 10 μl of 5 M ammonium acetate and 1 ml of absolute ethanol. After mixing by inversion, the tubes were centrifuged for 3 min at 13,500 rpm and the supernatant was discarded. Finally, the air-dried pellet was resuspended in 100 μl of TE buffer. To estimate the integrity of the isolated genomic DNA, 2.5 μl of the preparation was migrated on a 1% agarose gel. The concentration of the genomic DNA was determined with a Qubit 2.0 fluorimeter (Invitrogen).

Libraries were generated from 1 μg of genomic DNA and high-throughput sequencing was performed on a HiSeq 3000 instrument (Illumina). Library preparation and Illumina sequencing were carried out by the Next Generation Sequencing (NGS) Platform of the University of Bern. The raw reads (paired-end reads of 150 bp) were processed according to the following procedure: after performing a quality check with FastQC v0.11.2 (https://www.bioinformatics.babraham.ac.uk/projects/fastqc/), all the reads were filtered for quality (minimum of 20), truncated to 100 bp with Sickle v1.29[54] and then mapped with BWA-MEM v0.7.10[55] to the S. cerevisiae reference genome R64-1-1.79 (strain S288C) obtained from Ensembl[56]. The SAM files were sorted and converted to BAM files with SAMtools v1.2[57]. Single-nucleotide variants (SNVs), as well as small insertions and deletions (Indels), were identified with SAMtools and BCFtools v1.27[57]. Variant annotation was added with SnpEff v4.3[58]. Then, variants were filtered with SnpSift[59] to retain homozygous variants that are not found in the parental control strain and that are not 'synonymous' or 'intergenic', leading to an annotated and curated Variant Call Format (VCF) file. Results were viewed with the Integrative Genomics Viewer (IGV) software[60]. Deletion of the CGR1 gene was verified visually using IGV. Our sequence analysis revealed 13 variants for the three genomes and unambiguously identified one single-nucleotide change within the RPF2 gene in each suppressor strain.

**Yeast affinity purification.** Two-step affinity purifications were performed with either N-terminally TAP–Flag- or C-terminally Flag–TEV–proteinA (FTpA)-tagged bait proteins. The respective yeast strains were grown in 2 l of YPD medium at 30 °C, harvested in the logarithmic growth phase, flash frozen in liquid nitrogen and stored at −80 °C. Where indicated in the figures, cultures were incubated in the presence of 0.5 mM auxin (3-indoleacetic acid, Sigma–Aldrich) for 120 min prior to harvesting the cells. Cell pellets were resuspended in 'lysis buffer' [50 mM Tris-HCl (pH 7.5), 100 mM NaCl, 5 mM MgCl₂, 0.05% NP-40, 1 mM DTT, supplemented with 1 mM PMSF, 1 × SIGMAFAST protease inhibitor (Sigma–Aldrich)], and cells were ruptured by shaking in a bead beater (Fritsch) in the presence of glass beads. Lysates were cleared by two subsequent centrifugation steps at 4 °C for 10 and 30 min at 5000 and 14,000 rpm, respectively. Supernatants were incubated with immunoglobulin G (IgG) Sepharose 6 Fast Flow beads (GE Healthcare) on a rotating wheel at 4 °C for 90 min. Beads were transferred into Mobicol columns (Mobitec) and, after washing with 10 ml of lysis buffer, cleavage with tobacco etch virus (TEV) protease was performed at 16 °C for 120 min. In a second purification step, TEV eluates were incubated with Flag agarose beads (ANTI-FlagM2 Affinity Gel, Sigma–Aldrich) for 60 min at 4 °C. After washing with 5 ml of lysis buffer, bound proteins were eluted with lysis buffer containing 300 μg/ml Flag peptide at 4 °C for 45 min. Buffer lacking NP-40 was used for the last purification step in samples used for cryo-EM. Flag eluates were analysed by SDS-PAGE on 4–12% polyacrylamide gels (NuPAGE, Invitrogen) with colloidal Coomassie staining (Roti-blue, Roth) or by western blotting with antibodies, as indicated in the respective figures. Uncropped gel and western blot images are shown in Supplementary Fig. 10.

**Cryo electron microscopy.** Cryo electron microscopy was performed for three different purifications: (1) rrs1Δ [YCplac111–RRS1] Arx1–FTpA Cgr1–HA–AID; -auxin. (2) cgr1Δ depleted: rrs1Δ [YCplac111–RRS1] Arx1–FTpA Cgr1–HA–AID; + auxin (2 h). (3) cgr1Δ depleted in the presence of the rrs1E102D suppressor: rrs1Δ [YCplac111–rrs1E102D] Arx1–FTpA Cgr1–HA–AID + auxin (2 h).

For each purification, Quantifoil holy carbon grids (R3/3, +2 nm carbon) were glow discharged at 2.2*10^-1 torr for 20 s. Then for each grid, 3.5 μl of sample concentrated to 1.8 $OD_{260}$/ml was applied and plunge frozen in liquid ethane using a vitrobot mark IV (FEI), operating at 5 °C and 90% humidity, blotting for 2 s after a 45 s incubation. For each sample 400 micrograph were recorded on a Tecnai Spirit (FEI) operating at 120 kV, equipped with a TEMCam F216 (TVIPS, Germany). Semi-automated micrograph acquisition was performed using the EM-Tools software suite (TVIPS, Germany).

**Image processing.** GCTF[61] was used to estimate the contrast transfer function parameters. Micrographs with a defocus in the range of 0.8–3.2 μm were used for further processing. Template free particle picking was performed with Gautomatch (http://www.mrc-lmb.cam.ac.uk/kzhang). All further image processing (classifications, refinements, and post processing) was performed using Relion-2.1[62], analogously for all data sets as described in the following. First, the particle sets were cleaned using reference free 2D classification to eliminate falsely picked particles. Then, a consensus refinement was performed using EMD-3199[24] as a reference. To address structural heterogeneity, multiple subsequent steps of alignment free 3D classification was performed. After every classification step, similar classes were joined and all remaining classes were refined and sorted to check for additional heterogeneity (see Supplementary Fig. 8). For the cgr1Δ depleted sample, all classification attempts failed to separate the particles into subsets with structurally distinguishable features, resulting in one final class.

**Western blotting.** Western blot analysis was performed using the following antibodies: anti-Nog1 antibody (1:5000), anti-Nog2 antibody (1:20,000), anti-Arx1 antibody (1:2000), anti-Rei1 antibody (1:10,000), anti-Nsa2 antibody (1:10,000), anti-Rlp24 antibody (1:2000), provided by Micheline Fromont-Racine, anti-Nug1 antibody (1:10,000), anti-Yvh1 antibody (1:4000), provided by Vikram Panse, anti-Nmd3 antibody (1:10,000), anti-Rpl10 antibody (1:10,000), provided by Arlen Johnson, anti-Rpl3 antibody (1:5000), provided by Jonathan Warner, anti-Rpl5 antibody (1:10,000), provided by John Woolford, anti-Nop7 antibody (1:50,000), provided by Bruce Stillman, anti-Rsa4 antibody (1:10,000), provided by Miguel Remacha, anti-Mrt4 antibody (1:1000), provided by Juan Pedro Ballesta, anti-Arc1 antibody (1:5000), raised in our lab, anti-HA antibody (1:10,000, Covance Research Products, MMS-101R), horseradish-peroxidase-conjugated anti-Flag antibody (1:15,000, Sigma–Aldrich, A8592), secondary horseradish-peroxidase-conjugated goat anti-rabbit antibody (1:2000, Bio-Rad-170-6515), secondary horseradish-peroxidase-conjugated goat anti-mouse antibody (1:2000, Bio-Rad-170-6516).

**Polysome profile analyses.** Cells expressing chromosomal C-terminal fusions of Cgr1 tagged with HA–AID (CGR1–HA–AID) were grown in YPD medium to early logarithmic growth phase. Prior to harvesting, cultures were incubated with 0.5 mM auxin for 120 min to induce proteasomal degradation of Cgr1–HA–AID or left untreated. Subsequently, 100 μg/ml cycloheximide was added and after incubation for 10 min on ice, cells were pelleted and washed once with lysis buffer [50 mM Tris-HCl (pH 7.5), 100 mM KCl, 12 mM MgCl₂, 100 μg/ml cycloheximide]. After resuspension in lysis buffer and cell lysis with glass beads, 6 $A_{260}$ units of the cell extracts were loaded onto 10–50% sucrose gradients [dissolved in 50 mM Tris-HCl (pH 7.5), 100 mM KCl, 12 mM MgCl₂] and centrifuged with a SW40 rotor (Beckman Coulter) at 39,000 rpm for 2 h 45 min at 4 °C. Gradients were analysed on a Foxy Jr. fraction collector (Teledyne ISCO) with continuous monitoring at 254 nm.

**Fluorescence microscopy.** Living yeast cells expressing GFP- or RFP-tagged proteins were grown to the logarithmic growth phase and imaged by fluorescence microscopy using a Zeiss Imager Z1 microscope. As indicated, auxin was added to a final concentration of 0.5 mM and cells were subsequently incubated for 120 min prior to imaging.

## Data availability

All relevant data supporting the findings of this study can be found in the results or the supplementary information section and are available from the corresponding authors upon request. All experiments were performed at least twice with similar outcome. Cryo-EM densities of maps 1-7 of the Arx1 particles have been deposited in the Electron Microscopy Data Bank and can be retrieved using the following accession codes, respectively: EMDB-0218, EMDB-0219, EMDB-0220, EMDB-0221, EMDB-0222, EMDB-0223, EMDB-0224.

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

## Acknowledgements

We would like to thank Marén Gnädig for excellent technical support. We are grateful to Micheline Fromont-Racine, Vikram Panse, Arlen Johnson, Jonathan Warner, John Woolford, Bruce Stillman, Miguel Remacha and Juan Pedro Ballesta for the generous gift of antibodies. The research was supported by the Deutsche Forschungsgemeinschaft DFG (grants HU363/10-5, HU363/12-1 to E.H.) and by the Swiss National Science Foundation (grants 31003A_156764 and 31003A_175547 to D.K.).

## Author contributions

M.T. and E.H. conceived the study. M.T. and V.M. constructed yeast strains and plasmids. M.T., V.M., L.K., L.F. and D.K. performed the experiments. M.T., V.M., L.K., L.F., R.B., D.K. and E.H. analysed the data and discussed the results. M.T., V.M. and E.H. wrote the manuscript and all authors commented on the manuscript.

## Additional information

**Competing interests:** The authors declare no competing interests.

