## [Peer Review File · Nature Communications]

Reviewers' comments:

Reviewer #1 (Remarks to the Author):

Previously, it had been found that 5S rRNA, which is transcribed separately from 35S pre-rRNA, is assembled onto yeast ribosomes in a conformation very different from that in mature ribosomes, and therefore rotates almost 180 degrees during late nuclear stages of large subunit assembly.

Surprisingly and interestingly, Ed Hurt's group previously showed that, although the presence of the gigantic Rea1 ATPase in pre-ribosomes is required for this large rearrangement of 5S RNA relative to the rest of the pre-ribosome, *rea1* mutants defective in ATP hydrolysis still carry out this structural transition. This led them to hypothesize that binding of Rea1 to pre-ribosomes helps create a complex RNP network that stabilizes the rotated state of 5S rRNA cf. the pre-rotated state.

Here, the Hurt group provides further support for this hypothesis, using an elegant genetic approach. They begin with the idea that the small helical protein Cgr1, which binds near 5S rRNA in pre-ribosomes before and after rotation, somehow stabilizes the rotated conformation, and that in its absence, cells grow extremely slowly due to lack of sufficient stabilization of the rotated 5S RNA conformation in pre-ribosomes. They demonstrate that spontaneous extragenic suppressors of the very slow growing *cgr1* null mutant result from mutations in the *RPF2*, *RRS1* or *RPL5* genes. These genes encode three proteins that form a network of interactions with the 5S rRNA in the un-rotated state, but less so in the rotated state in part due to the exit of Rpf2 and Rrs1 prior to rotation.

I have only minor quibbles with the quality of the data presented (see below). However, this manuscript could be significantly improved by addressing at least two issues:

(1) I believe that that the manuscript as written places too much emphasis on Cgr1. The results do not prove that Cgr1 is a "key regulator of 5S RNP rotation", as stated in the title. Indeed, the authors appropriately use very careful wording elsewhere, such as these results "suggest" or "hint".... I think a better description of this work, in the title and elsewhere in the manuscript, should focus on the roles of Rpf2, Rrs1, and rpl5. It seems odd that Cgr1 is not totally essential but is touted as a "key regulator". That being said, we all struggle with the difference between "almost essential" and "essential", when describing such proteins.

(2) Second, the data presented are consistent with the present model for approximately when in the assembly pathway, and where in the cell, 5S RNP rotation occurs. The results presented show that in the *cgr1* null mutant, assembly is blocked before exit of Rpf2, Nug2, and Rsa4, and before nuclear export. Nevertheless, it is disappointing that the authors show no cryo-EM structural data to directly assay whether the 5S RNP has not rotated in the *cgr1* mutant, but rotated in the suppressor mutant strains.

Minor issues:

(1) Introduction: why do the authors say there are approximately 45 ribosomal proteins? Is there an agreed-upon number for how many ribosomal proteins are present in yeast ribosomes?

(2) Figure 1A: please show helix 38 in this diagram of the pre-ribosome structure. Rearrangement of this helix during 5S rRNA rotation seems to play a key role. Also, please show a better-oriented view of the particle to better illustrate all of the points made.

(3) Fig.1A: please illustrate where in the 3D structure are the *cgr1* mutations used.

(4) Fig. 2B: why are there halfmer polyribosomes in the wild type sample?

(5) Fig. 2D: I think that the Arx1 affinity-purified sample from the Cgr1 depleted strain is under-loaded, since there is less Arx1 present than in the un-depleted strain. This might partly explain why there is considerably less Nop7 in the depleted sample, as well as less Nmd3, Rei1, etc. Nevertheless, it is still evident that levels of proteins that are normally released from pre-ribosomes after 5S rRNA rotation go way up in the depleted sample, which is consistent with the model proposed!

(6) The authors indicate that levels of Rpf2 increase in pre-ribosomes from the Cgr1 depleted strain, based on the intensity of the stained band. Yet, they indicate in Fig. 5B that there are two other proteins in this band, Rlp7 and Yvh1. Thus, without western blotting or quantitative mass spec, how can they make this conclusion? (even though other results are consistent with this idea). And, the authors should use consistent labeling for this band or these bands in Fig. 2E cf. Fig. 5B.

(6) Fig. 2E and supplementary Fig. 2: why is the growth of the *cgr1*ΔN51 strain different in several otherwise wild-type strains (NSA2 and NOP7 cf. NUG1 and RIX1)?

(7) In order to make this manuscript more accessible to the general readership (non yeast geneticists), I suggest that the authors add to the Supplemental section a brief but clear step by step description of exactly how they constructed the double-shuffle yeast strains.

(8) Supplementary Table 2: please proofread again; there are a number of tiny typos: the "R" in RPF2 is not italicized. Also, does the P that is sometimes present upstream of RPF2 refer to the RPF2 promoter???

(9) Just a question: I wonder whether *cgr1* knockouts are NOT slow growing in some otherwise "wild-type" strains, simply due to the efficiency of acquisition of suppressors of the knockout. Perhaps one could sequence the RPF2, RRS1, and RPL5 genes from these strains.

Respectfully,
John Woolford

Reviewer #2 (Remarks to the Author):

Based on cryo-EM structures of pre-60S maturation intermediates it was previously suggested that the C-terminus of 60S-bound Cgr1 undergoes a large conformational change from a bent helix to a straightened form (Wu et al, 2016). In this work the authors now perform biochemical and genetic analysis to reveal the function of Cgr1 in 60S biogenesis. To that end they employ yeast strains, in which CGR1 is deleted, or which are depleted from Cgr1 via the AID/auxin system. The data suggest that Cgr1 may play a role in the proper assembly/maturation of the 5S rRNA.

As detailed below I suggest some controls and improved data presentation (e.g. with respect to the characterization of the different pre-60S complexes). In addition it would greatly improve the impact of the study if structural data in support of the suggested model for Cgr1-function could be included. This seems feasible as the Beckman group is part of the team. It seems logical to determine the structure of Arx1 pre-60S particles with or without Cgr1 (as shown in Fig. 2d), to confirm that indeed Cgr1 affects the rotational state of 5S rRNA in pre-60S particles. (... not only to present models based on preexisting structures ...)

I feel this is important also because of the informative, saturated suppressor screen shown in Fig. 3.

This screen identified Rpf2, Rrs1, and Rpl5 mutants as the only extragenic suppressors of the Δ cgr1 deletion. Based on these data the authors suggest that the suppressor mutations may "drive the equilibrium towards the rotated state of the 5S RNP" and by that compensate for the absence of Cgr1. This may be the case. However, I feel an additional model may also explain the data. It was previously shown that Rpl5 assembles with 5S rRNA prior to incorporation into the ribosome and Rpf2-Rrs1 interacts with 5S/Rpl5 prior to the incorporation of the 5S/Rpl5 RNP into pre-60S particles (Madru et al, 2015). The results of the suppressor screen are compatible with a model in that Cgr1 facilitates the initial Rpf2/Rrs1-dependent incorporation of the 5S RNP into pre-60S. In this alternative model, the suppressor mutations in Rpl5, Rpf2, or Rrs1 may facilitate the assembly of the 5S RNP in the absence of Cgr1.

One simple experiment to distinguish between the two possibilities is to test for the presence/amount of Rpl5 (and also 5S rRNA) in Arx1 pre-60S particles with or without Cgr1 (see Fig. 2d; I am a bit surprised that the authors did not perform this control, as the Rpl5 antibody was employed in Fig. 1d).

In addition, visualization of pre-60S particles via cryo EM shall reveal if the conformation of the 5S rRNA is altered depending on the presence of Cgr1 as suggested (see above).

Additional major points:

Fig 1d, Fig. 1e, Fig. 2d, Fig. 5b

The authors state that "bands were identified by mass spectrometry". I am missing the data/results/controls of this important experimental part of the study. Why were only some of the proteins in the purified complexes identified?

With respect to the immunoblots of Fig 1d, Fig. 1e, Fig. 2d, Fig. 5b: Please show input controls (fraction of the total material, which was used for the affinity purifications). Importantly, the analysis should include immunoblots for a small ribosomal subunit protein to reveal possible contamination of affinity purified pre-60S particles with assembled ribosomes.

Fig 1e and Fig. 2b

Does the strain expressing Cgr1-3xHA display wild type growth? Please show a growth control (plating). Was Cgr1 tagged C-terminally? This was not clear to me from the description. Seems critical. Please show that the 3xHA does not interfere with ribosome-binding. Same with respect to the CGR1-HA-AID construct (see below). The ribosome profile shown in Fig. 2b suggests an assembly defect already before auxin treatment.

Fig. 2b

The fractions of the ribosome profile should be analyzed for CGR1-HA-AID to confirm that the tagged protein is properly assembled into ribosomes (see also above).

Fig. 5b

In this experiment Rpl5 is labeled on the Coomassie gel. However, quantification of the amount of Rpl5 on the Coomassie gel is not possible. To that end, please show an Rpl5 blot in addition to the Rpl3 blot (see also above).

Additional points.

Fig. 1c

Please show a nuclear stain to facilitate visual localization of GFP-Cgr1.

Fig. 1e

Why is the size of Nug2 shifted in the Rix1- and Arx1 pre-60S particles?

Fig. 2b

Halfmer polysomes are diagnostic for 48S initiation complexes on actively translated mRNAs. One possible (but not the only) reason for this situation is a 60S biogenesis defect. Please state clearly in the Figure Legend and Results. Please also reference (Moy et al, 2002) who previously showed the halfmer phenotype of cells lacking Cgr1.

Fig 2c

What is shown in the lowest panel of the Fig. 2c? Probably this is not CGR1-HA-AID cells?

Discussion

"This rearranged topology suggests that Cgr1 accompanies or even drives H38 relocation from the solvent to the inter-subunit side." - Some speculation how that may work?

Discussion of the p53 connection at the end of the Discussion seems somewhat detached from the the rest of the manuscript.

Comments to the reviewer

Reviewer #1 (Remarks to the Author):

Previously, it had been found that 5S rRNA, which is transcribed separately from 35S pre-rRNA, is assembled onto yeast ribosomes in a conformation very different from that in mature ribosomes, and therefore rotates almost 180 degrees during late nuclear stages of large subunit assembly. Surprisingly and interestingly, Ed Hurt's group previously showed that, although the presence of the gigantic Rea1 ATPase in pre-ribosomes is required for this large rearrangement of 5S RNA relative to the rest of the pre-ribosome, *rea1* mutants defective in ATP hydrolysis still carry out this structural transition. This led them to hypothesize that binding of Rea1 to pre-ribosomes helps create a complex RNP network that stabilizes the rotated state of 5S rRNA cf. the pre-rotated state.

Here, the Hurt group provides further support for this hypothesis, using an elegant genetic approach. They begin with the idea that the small helical protein Cgr1, which binds near 5S rRNA in pre-ribosomes before and after rotation, somehow stabilizes the rotated conformation, and that in its absence, cells grow extremely slowly due to lack of sufficient stabilization of the rotated 5S RNA conformation in pre-ribosomes. They demonstrate that spontaneous extragenic suppressors of the very slow growing *cgr1* null mutant result from mutations in the *RPF2*, *RRS1* or *RPL5* genes. These genes encode three proteins that form a network of interactions with the 5S rRNA in the un-rotated state, but less so in the rotated state in part due to the exit of Rpf2 and Rrs1 prior to rotation.

I have only minor quibbles with the quality of the data presented (see below). However, this manuscript could be significantly improved by addressing at least two issues:

(1) I believe that that the manuscript as written places too much emphasis on Cgr1. The results do not prove that Cgr1 is a “key regulator of 5S RNP rotation”, as stated in the title. Indeed, the authors appropriately use very careful wording elsewhere, such as these results “suggest” or “hint”.... I think a better description of this work, in the title and elsewhere in the manuscript, should focus on the roles of Rpf2, Rrs1, and rpl5. It seems odd that Cgr1 is not totally essential but is touted as a “key regulator”. That being said, we all struggle with the difference between “almost essential” and “essential”, when describing such proteins.

According to this suggestion, we changed the title of the manuscript to “**Suppressor mutations in the Rpf2-Rrs1 module or Rpl5 bypass the Cgr1 function for pre-ribosomal 5S RNP rotation**” and also avoid the term key regulator in the text.

(2) Second, the data presented are consistent with the present model for approximately when in the assembly pathway, and where in the cell, 5S RNP rotation occurs. The results presented show that in the *cgr1* null mutant, assembly is blocked before exit of Rpf2, Nug2, and Rsa4, and before nuclear export. Nevertheless, it is disappointing that the authors show no cryo-EM structural data to directly assay whether the 5S RNP has not rotated in the *cgr1* mutant, but rotated in the suppressor mutant strains.

As suggested by this reviewer, we have performed cryo-EM and analyzed in direct comparison 3 different Arx1-affinity purified pre-60S particles:

- (i) Cgr1⁺ control: *rrs1*Δ [YCplac111-RRS1] Arx1-FTpA Cgr1-HA-AID - Auxin
- (ii) *cgr1* depleted: *rrs1*Δ [YCplac111-RRS1] Arx1-FTpA Cgr1-HA-AID + Auxin (2h)
- (iii) *cgr1* depleted in the presence of the Rrs1E102D suppressor: *rrs1*Δ [YCplac111-*rrs1*E102D] Arx1-FTpA Cgr1-HA-AID + Auxin (2h)

This EM analysis revealed that the 5S RNP remained in the 5S RNP non-rotated state to 100% in the *cgr1* depletion mutant, whereas in the *Cgr1* non-depleted strain (control) both non-rotated (~60%) and rotated (~40%) forms were found (see revised Fig. 6). This latter ratio roughly corresponds to the published early Arx1 or Nog2 state pre-60S particles (Leidig et al., 2014, Wu et al., 2016). Of further interest here is that the foot structure was absent from the *Cgr1*-depleted particles, suggesting that maturation of the foot can proceed uncoupled of the 5S RNP maturation. Notably, the Arx1 particles in the *rrs1E102D* suppressor strain showed a significant portion of 5S RNP relocated particles (23%), suggesting that the suppressor mutation has facilitated the 5S RNP rotation even without *Cgr1*, which also could explain why the suppressor strain has improved growth properties and restored nuclear export of pre-60S particles.

Minor issues:

(1) Introduction: why do the authors say there are approximately 45 ribosomal proteins? Is there an agreed-upon number for how many ribosomal proteins are present in yeast ribosomes?

We now give the exact number for yeast and human ribosomal proteins.

(2) Figure 1A: please show helix 38 in this diagram of the pre-ribosome structure. Rearrangement of this helix during 5S rRNA rotation seems to play a key role. Also, please show a better-oriented view of the particle to better illustrate all of the points made.

We now show a better-oriented view and depict helix 38 in the revised Fig. 1a.

(3) Fig. 1A: please illustrate where in the 3D structure are the *cgr1* mutations used.

We cannot exactly indicate these mutations in the *Cgr1* model shown in Fig. 1a, for which no pdb exists (Rix1-Rea1 particle, PDB 5JCS). Instead, we have now labeled the *cgr1RRR(108,109,110)>AAA* and *cgr1ΔN51* mutations in the corresponding *Cgr1* structure of the Nog2 particle (PDB 3JCT), in which the 5S RNP is still in the non-rotated stage (revised Supplementary Fig. 3b).

(4) Fig. 2B: why are there halfmer polyribosomes in the wild type sample?

It appears that the HA-AID fused to the C-terminus of *Cgr1* could be the reason for this slight halfmer phenotype, despite the fact that the strain does not display an obvious growth defect (see Supplementary Fig. 2). Moreover, we do not see an apparent accumulation of the Rpl25-GFP reporter in the *Cgr1*-AID-HA strain before auxin induced depletion. Taken all these observations together, the AID-degron attached to *Cgr1* may affect mildly the *Cgr1* function, which we now mention in the text. However, it is clear that depletion of *Cgr1* severely increases the halfmer phenotype with a strong nuclear accumulation of the Rpl25-GFP reporter.

(5) Fig. 2D: I think that the Arx1 affinity-purified sample from the *Cgr1* depleted strain is under-loaded, since there is less Arx1 present than in the un-depleted strain. This might partly explain why there is considerably less Nop7 in the depleted sample, as well as less Nmd3, Rei1, etc. Nevertheless, it is still evident that levels of proteins that are normally released from pre-ribosomes after 5S rRNA rotation go way up in the depleted sample, which is consistent with the model proposed!

Indeed, in the previous gel the sample derived from the Cgr1-depleted strain was slightly under-loaded. We have repeated this experiment with adjusted loading, and also used a Rpf2-3xHA strain as suggested by this reviewer to follow Rpf2 levels directly by western blotting (revised Fig. 2d). This all now clearly reveals that the levels of Nop7, Yvh1, and Nmd3 are reduced on particles lacking Cgr1. For Yvh1 and Nmd3 this result is consistent with joining of these factors being hindered by the arrest of pre-60S maturation at a stage before 5S RNP rotation. The strong decrease of Nop7 from such particles can be explained by the observation that foot factor removal is independent of 5S RNP rotation (see above), which is also consistent with Coomassie staining (see Fig. 2d and Fig. 5b), showing that also the foot-factor Cic1/Nsa3 is reduced on the particle upon Cgr1 depletion.

(6) The authors indicate that levels of Rpf2 increase in pre-ribosomes from the Cgr1 depleted strain, based on the intensity of the stained band. Yet, they indicate in Fig. 5B that there are two other proteins in this band, Rlp7 and Yvh1. Thus, without western blotting or quantitative mass spec, how can they make this conclusion? (even though other results are consistent with this idea). And, the authors should use consistent labeling for this band or these bands in Fig. 2E cf. Fig. 5B.

We constructed Rpf2 tagged with 3xHA, which allowed Rpf2 detection by western blotting in the Arx1-FTpA Cgr1-HA-AID strain. Affinity-purifications and western blot analyses from this strain revealed a distinct western signal increase of Rpf2-3xHA on Cgr1-depleted particles (see revised Figure 2d). Moreover, the Rpf2-3xHA (that due to the tag now migrates slower than Yvh1) co-enrichment is also visible on the Coomassie stained gel. These findings, together with western blot analyses, showing that Yvh1 is absent from Cgr1-depleted particles, further supports our earlier conclusion that Rpf2 becomes enriched Cgr1-depleted Arx1 particles (Figure 5b).

As suggested, we now use a consistent labelling for the Coomassie stained bands in Fig. 2d and Fig. 5b

(6) Fig. 2E and supplementary Fig. 2: why is the growth of the *cgr1*ΔN51 strain different in several otherwise wild-type strains (NSA2 and NOP7 cf. NUG1 and RIX1)?

From Supplementary Fig. 2, where cells are spotted on YPD plates, it is evident that the *cgr1*ΔN51 allele causes a temperature-sensitive phenotype, whereas the strain displays only a mild growth defect at 23°C and 30°C. In Fig. 2e, cells were spotted on 5-FOA plates and incubated at 30°C. 5-FOA medium allows a first estimation and comparison of cellular growth and viability, but is not suitable to assess growth rates in a precise way. We assume that the *cgr1*ΔN51 allele in the respective wild-type strains (*NSA2*, *RIX1*, *NUG1*, *NOP7*) that were incubated at 30°C (shown in Fig. 2e) also displays a similar mild growth defect as observed for *cgr1*ΔN51 in the wild-type strain spotted on YPD and incubated at 30°C (Supplementary Fig. 2).

(7) In order to make this manuscript more accessible to the general readership (non yeast geneticists), I suggest that the authors add to the Supplemental section a brief but clear step by step description of exactly how they constructed the double-shuffle yeast strains.

We included a description of how the double shuffle strains were constructed in the methods section.

(8) Supplementary Table 2: please proofread again; there are a number of tiny typos: the “R”

in RPF2 is not italicized. Also, does the P that is sometimes present upstream of RPF2 refer to the RPF2 promoter???

We have corrected these typos. P denotes promoter, which is now indicated as a footnote.

(9) Just a question: I wonder whether *cgr1* knockouts are NOT slow growing in some otherwise "wild-type" strains, simply due to the efficiency of acquisition of suppressors of the knockout. Perhaps one could sequence the RPF2, RRS1, and RPL5 genes from these strains.

This is an interesting possibility that such strains may carry polymorphic changes in the *RPF2*, *RRS1*, and *RPL5* genes. However, since we do not have these other strains currently available in our lab, we plan to perform this experiment in future studies.

Reviewer #2 (Remarks to the Author):

Based on cryo-EM structures of pre-60S maturation intermediates it was previously suggested that the C-terminus of 60S-bound Cgr1 undergoes a large conformational change from a bent helix to a straightened form (Wu et al, 2016). In this work the authors now perform biochemical and genetic analysis to reveal the function of Cgr1 in 60S biogenesis. To that end they employ yeast strains, in which *CGR1* is deleted, or which are depleted from Cgr1 via the AID/auxin system. The data suggest that Cgr1 may play a role in the proper assembly/maturation of the 5S rRNA.

As detailed below I suggest some controls and improved data presentation (e.g. with respect to the characterization of the different pre-60S complexes). In addition it would greatly improve the impact of the study if structural data in support of the suggested model for Cgr1-function could be included. This seems feasible as the Beckman group is part of the team. It seems logical to determine the structure of Arx1 pre-60S particles with or without Cgr1 (as shown in Fig. 2d), to confirm that indeed Cgr1 affects the rotational state of 5S rRNA in pre-60S particles. (... not only to present models based on preexisting structures ...)

I feel this is important also because of the informative, saturated suppressor screen shown in Fig. 3. This screen identified Rpf2, Rrs1, and Rpl5 mutants as the only extragenic suppressors of the $\Delta cgr1$ deletion. Based on these data the authors suggest that the suppressor mutations may "drive the equilibrium towards the rotated state of the 5S RNP" and by that compensate for the absence of Cgr1. This may be the case. However, I feel an additional model may also explain the data. It was previously shown that Rpl5 assembles with 5S rRNA prior to incorporation into the ribosome and Rpf2-Rrs1 interacts with 5S/Rpl5 prior to the incorporation of the 5S/Rpl5 RNP into pre-60S particles (Madru et al, 2015). The results of the suppressor screen are compatible with a model in that Cgr1 facilitates the initial Rpf2/Rrs1-dependent incorporation of the 5S RNP into pre-60S. In this alternative model, the suppressor mutations in Rpl5, Rpf2, or Rrs1 may facilitate the assembly of the 5S RNP in the absence of Cgr1.

One simple experiment to distinguish between the two possibilities is to test for the presence/amount of Rpl5 (and also 5S rRNA) in Arx1 pre-60S particles with or without Cgr1 (see Fig. 2d; I am a bit surprised that the authors did not perform this control, as the Rpl5 antibody was employed in Fig. 1d).

Actually, we did not perform Rpl5 western blots in the first round, since Coomassie staining of the Arx1 eluates in presence or absence of Cgr1 clearly showed similar amounts of Rpl5 on both particles. As suggested, in the new purifications shown in the revised Figure 2d, we now did western blotting using an Rpl5 antibody, which confirmed that Rpl5 levels on Arx1 particles purified from wild-type or from Cgr1-depleted cells are similar. This result suggests that the recruitment of the 5S RNP to the pre-60S particle does not depend on Cgr1 and is consistent with other data showing that Cgr1 is not present on early nucleolar pre-60S particles (Ssf1 and Nsa1; see Fig. 1e), in contrast to the 5S RNP, which is recruited already at the stage of nucleolar precursors (Kater et al, 2017; Sanghai et al., 2018; Zhou et al., 2018)

In addition, visualization of pre-60S particles via cryo EM shall reveal if the conformation of the 5S rRNA is altered depending on the presence of Cgr1 as suggested (see above).

See also our response to reviewer #1. We performed cryo-EM analyses, which clearly revealed that the 5S RNP is not rotated on all Arx1 particles isolated from the strain lacking Cgr1, whereas wild-type and Cgr1-depleted particles but isolated from a suppressor mutant showed a significant 5S RNP rotation (revised Fig. 6).

Additional major points:

Fig 1d, Fig. 1e, Fig. 2d, Fig. 5b

The authors state that "bands were identified by mass spectrometry". I am missing the data/results/controls of this important experimental part of the study. Why were only some of the proteins in the purified complexes identified?

The fast migrating bands at the bottom of the gel largely represent ribosomal proteins well-known to associate with pre-60S particles, thus we did not subject these bands to MS analyses.

With respect to the immunoblots of Fig 1d, Fig. 1e, Fig. 2d, Fig. 5b: Please show input controls (fraction of the total material, which was used for the affinity purifications). Importantly, the analysis should include immunoblots for a small ribosomal subunit protein to reveal possible contamination of affinity purified pre-60S particles with assembled ribosomes.

We now included western blot analyses of the respective lysates, which were used to affinity purify the particles shown in the revised Figure 2d. In addition, we analysed Rps3 in the lysates and eluates by Western blotting, which can be routinely detected by Western in TAP purifications due to the stickiness of mature ribosomes in TAP eluates. However, the amount of contamination was very similar between the wild-type and Cgr1-depleted samples.

Fig 1e and Fig. 2b

Does the strain expressing Cgr1-3xHA display wild type growth? Please show a growth control (plating). Was Cgr1 tagged C-terminally? This was not clear to me from the description. Seems critical. Please show that the 3xHA does not interfere with ribosome-binding. Same with respect to the CGR1-HA-AID construct (see below). The ribosome profile shown in Fig. 2b suggests an assembly defect already before auxin treatment.

As requested, we performed growth analyses with the C-terminally 3xHA-tagged Cgr1 strain (see new Supplementary Fig. 1). In addition, we analysed growth of two strains used for affinity purifications shown in Fig. 1e (i.e. Cgr1-3xHA Nug2-FtpA and Cgr1-3xHA Rix1-FTpA)

and in Fig. 1d (i.e. TAP-Flag-Cgr1 (N-terminal fusion) and Cgr1-FTpA (C-terminal fusion)). All analysed strains display a growth, which is similar to the wild-type strain. Furthermore, in Supplementary Fig. 2, the growth analysis of the C-terminally tagged Cgr1-HA-AID is shown. Also this strains displays a wild-type growth, yet we observe a slight half-mer phenotype in the in the polysome profile, which we attribute to the bulky HA-AID degron tag (see also our response to reviewer #1, minor issue 4).

Fig. 2b

The fractions of the ribosome profile should be analyzed for CGR1-HA-AID to confirm that the tagged protein is properly assembled into ribosomes (see also above).

From the affinity-purifications of N- and C-terminally tagged Cgr1 as bait protein (Fig. 1d), as well as from detection of Cgr1-3xHA tagged (Fig. 1e) and Cgr1-HA-AID (Fig. 2d and Fig. 5b) on pre-60S particles, in combination with the now performed growth controls (see above), all tagged versions appear to be fully functional with normal incorporation into pre-ribosomal particles.

Fig. 5b

In this experiment Rpl5 is labeled on the Coomassie gel. However, quantification of the amount of Rpl5 on the Coomassie gel is not possible. To that end, please show an Rpl5 blot in addition to the Rpl3 blot (see also above).

In the revised Figure 2d, we now show by western blot detection of Rpl5 that the protein is efficiently recruited also to Cgr1-depleted pre-60S particles. Since the western blot in Fig. 2d is in good agreement with the Rpl5 amounts stained by Coomassie, we suggest that also in Fig. 5b the Rpl5 levels are similar in the respective purifications, as seen in the Coomassie gel.

Additional points.

Fig. 1c

Please show a nuclear stain to facilitate visual localization of GFP-Cgr1.

As suggested, we performed fluorescence microscopy with RFP-Nop1 as nucleolar marker and observed co-localization of GFP-Cgr1 with Nop1 within the nucleolus. Indicated by some non-overlapping GFP signal in the nucleus, Cgr1 also localizes to the nucleoplasm, which is consistent with the biochemical data.

Fig. 1e

Why is the size of Nug2 shifted in the Rix1- and Arx1 pre-60S particles?

After Nug2-FTpA purification, Nug2 still contains the Flag-tag in the final eluate (only the ProtA tag is cleaved off). Hence, Nug2-Flag migrates slower than untagged Nug2 present in the Arx1 and Rix1 eluates.

Fig. 2b

Halfmer polysomes are diagnostic for 48S initiation complexes on actively translated mRNAs. One possible (but not the only) reason for this situation is a 60S biogenesis defect. Please state clearly in the Figure Legend and Results. Please also reference (Moy et al, 2002) who previously showed the halfmer phenotype of cells lacking Cgr1.

We adjusted the figure legend and refer to Moy et al, 2002 in the results.

Fig 2c

What is shown in the lowest panel of the Fig. 2c? Probably this is not CGR1-HA-AID cells?

The lowest panel shows the localization of Rps3-GFP after Cgr1 depletion (CGR1-HA-AID cells + Auxin). The nuclear export of this 40S reporter is not inhibited, thus the defect after Cgr1 depletion is specific for the large subunit.

Discussion

"This rearranged topology suggests that Cgr1 accompanies or even drives H38 relocation from the solvent to the inter-subunit side." - Some speculation how that may work?

We have given some speculation in the Discussion of the manuscript to give a hint how H38 may be relocated by the action of Cgr1.

Discussion of the p53 connection at the end of the Discussion seems somewhat detached from the rest of the manuscript.

We see this point, and accordingly have removed this last paragraph from the Discussion.

REVIEWERS' COMMENTS:

Reviewer #1 (Remarks to the Author):

The authors have very thoroughly and nicely addressed all of my concerns!

John Woolford